# Implementation fidelity of school oral health programs at a District in South Africa

Mpho Molete[1]☯*, Aimee Stewart[2]☯, Jude Igumbor[3]‡

**1** Department of Community Dentistry, University of the Witwatersrand, School of Oral Health Sciences, Johannesburg, South Africa, **2** University of the Witwatersrand, School of Therapeutic Sciences, Johannesburg, South Africa, **3** University of the Witwatersrand, School of Public Health, Johannesburg, South Africa

☯ These authors contributed equally to this work.
‡ These author also contributed equally to this work.
* Mpho.molete@wits.ac.za

## Abstract

### Background

It is important that components contributing to success of a program are well understood to ensure better outcomes and strengthen interventions. Hence the purpose of the study was to assess the level of fidelity achieved by school oral health programs in our study district and to determine elements of fidelity that predict the risk of dental decay.

### Methods

A cross-sectional study design was utilised. A multistage sampling technique was employed to randomly select 10 schools, two grades in each school were selected and all pupils in the selected grades were included in an oral health examination. Ten oral hygienists were observed and interviewed as they carried out the activities of the program and records were reviewed. Data collection tools included an oral health examination form, and an implementation fidelity checklist.

### Results

The average level of fidelity obtained was 40% and it was shown to be inversely correlated with levels of decay, as decay was predicted to decrease with increasing levels of fidelity. The fidelity elements that were found to directly predict the outcome of decay included duration (IRR, 0.49; p = 0.02) coverage (IRR, 0.54; p = 008), content (IRR, 1.36; p = 0.03) and age (IRR, 2.14; p = 0.00). Moderating factors of fidelity which indirectly influenced the outcome of decay included facilitation strategy, duration and age. These were predicted to reduce the risk of decay by 92%, 83% and 48% respectively.

### Conclusion

The school oral health programs exhibited high levels of pupil coverage, however, the content of the programs offered was low (28%). Coverage was high in the context of lack of

**Data Availability Statement:** All relevant data are within the manuscript and its supporting information files.

**Funding:** The First Author (MM) is funded by the following for a PhD program. The Consortium for

Advanced Research Training in Africa (CARTA), this is jointly led by the African Population and Health Research Centre and the University of the Witwatersrand and funded by the Carnegie Corporation of New York (Grant No–B 8606.R02), Sida (Grant No:54100113), the DELTAS Africa Initiative (Grant No: 107768/Z/15/Z) and Deutscher Akademischer Austauschdienst (DAAD). The DELTAS Africa Initiative is an independent funding scheme of the African Academy of Sciences (AAS)'s Alliance for Accelerating Excellence in Science in Africa (AESA) and supported by the New Partnership for Africa's Development Planning and Coordinating Agency (NEPAD Agency) with funding from the Wellcome Trust (UK) and the UK government. Other Funders: National Research Foundation (NRF Thuthuka) and the Health & Welfare Sector Education & Training Authority (HWSETA) in South Africa. The funders had no role in the study design, data collection and analysis, decision to publish, or preparation of the manuscript.

**Competing interests:** The authors have declared that no competing interests exist.

dental assistance and time. Multi-sectoral participation is therefore necessary to re-organise the program for improving implementation fidelity and bringing about quality implementation.

## Background

School health programs date back to the 1950's with the aim of facilitating collaboration between the educational and health sectors in working towards improving education and health outcomes of children in one setting [1]. The philosophy behind such programs has been further shaped by the Ottawa Charter on health promotion in terms of addressing conditions affecting school children by building healthy school policies; creating supportive environments; strengthening community actions; developing personal skills and re-orientation towards disease prevention and health promotion [2, 3].

Oral health has often been one of the first activities to be adopted by schools as it can be incorporated into key elements of health promotion and provides an opportunity for addressing oral health within a general health promotion strategy [2]. Oral health in South Africa is also included in the Integrated School Health Policy which was legislated in 2012. The aim of the policy is to improve primary health care delivery, children's general health and education outcomes [4]. One of the pilot sites for integrated school health is in the study district of Tshwane where oral health services offered include; oral health screening, education, brushing programs and fissure sealants. These programs have been officially offered since 2011 however they have not been assessed for implementation effectiveness [4–6].

School oral health programs in Tshwane are offered by the district oral hygienists who all have a qualification in Oral Hygiene that include a module on the implementation of community programs. The oral hygienists are employed by the Department of Health in the Gauteng province; they are based in local district dental clinics and are responsible for community and clinical preventative oral health care. In the school settings the hygienists offer oral health education and co-ordinated tooth brushing programs to groups of children. In addition they use portable dental equipment that is set up to provide, oral health screening, fissure sealants and simple restorations such as the Atraumatic Restorative Technique (ART).

The oral hygienists spend approximately three to four weeks of four hours per morning at a specific school delivering the program. They are expected to complete a school after attending to at least one primary grade class and one secondary grade class. At the schools they offer mass oral health education to the teachers and all children, co-ordinated tooth brushing activities, screening, fissure sealants and ART to the children who require the treatment. The class teachers are involved in supervising the children on the tooth brushing program after lunch times. Upon completing a specific school, the oral hygienists are expected to return to the school quarterly during the year to reinforce the education and the brushing activities.

Data from a global school oral health survey and a review of school health services, found that the full range of health promotion at schools was narrowly implemented and not applied broadly as recommended by the Ottawa Charter [7, 8]. Furthermore studies have shown that although over 80% of the documented programs reported positive oral health outcomes as a result of interventions, there was general poor reporting of the extent of implementation [9–13].

It is recommended that interventions should be monitored through stages of design, development and implementation in order that accurate interpretations of how elements of the interventions contribute to the desired outcomes can be made [14]. Such deductions are made possible by the concept of fidelity, which refer to the application of an intervention as

originally designed. In understanding how the intervention was designed and the operational processes leading to its success, evidence can then be documented on how to strengthen an entire intervention or to precisely target poor performing components of the intervention. [15]. A program with high fidelity would therefore be considered to produce favorable outcomes [16, 17].

Implementation fidelity is not a linear construct as it is influenced by various contextual factors and its measurement is complex [17].There are various frameworks of fidelity and our study adopted Carrol's framework as it allows for measurements of the active ingredients of an intervention and factors that influence the quality of implementation [17, 18]. Elements of fidelity as described by Carrol (2007) include adherence and potential moderators. Adherence refer to content, coverage, frequency and duration, these express the level of implementation fidelity achieved. Potential moderators are elements which are suggested to have some influence on the level of implementation achieved, these include; intervention complexity, facilitation strategies, quality of delivery and participant responsiveness [19].

School oral health programs have been in existence for over 20 years in South Africa however the oral health status of school pupils in South Africa remains poor with persistently high levels (80%) of unmet treatment needs [11, 20]. There is also no evidence of monitoring and evaluation of school oral health by the Department of Health in South Africa. Therefore conducting this analysis may provide insight into fidelity elements affecting the dental decay outcome of the school pupils receiving school oral health programs.

The objectives of our study were to assess the level of fidelity achieved by the district school oral health programs; to determine how adherence elements of fidelity predict the risk of decay, and how potential moderating factors of fidelity predict the risk of decay.

## Methodology

This cross-sectional study was conducted in the study district situated in the northern area of the Gauteng Province, South Africa. A multistage sampling technique was undertaken to select both schools and learners. There were 10 oral hygienists in the district providing oral health services and they were all invited to participate. Each of the oral hygienists had a list of schools receiving oral health programs; therefore one school from each of the hygiene list was randomly selected for participation. The schools selected for participation had to have been receiving the oral health program for more than one year.

In the first stage of sampling, one school was randomly selected from each oral hygienist list. Upon selecting the 10 schools, two grades in each school were selected to include children with ages ranging between 6 and 7 and 12 and 13 years as these are the ages recommended for dental programs in the National Oral Health Policy [21]. The last stage of sampling included a random selection of two classes within the chosen grades, which were grades 1 and 7's at each participating school. All the children in the selected classes were invited to participate in an oral health examination.

About 700 pupils were required to enable the estimation of the mean Decayed Missing Filled Teeth (DMFT) with a precision of ± 0.3, assuming the standard deviation was 2.0 with a design effect of 2. The calculation also took into account the clustering effect of pupils within schools (Statcalc, version 14).

Pupils were invited to have oral health examinations which were undertaken using the DMFT index, which provides information on the decay levels of the permanent teeth. All information was captured on an oral health assessment form and procedures were followed according to WHO guidelines [22]. The first author and a research assistant conducted the oral health examinations. They were calibrated by a dental public health specialist not involved

in the research and inter-rater reliability was established by conducting repeat examination of images up until the Cohen's Kappa score was ≥ 0.80.

The oral hygienists were also observed and interviewed as they carried out the activities of the program. Records were also reviewed to verify some information and the data collection was guided by an implementation fidelity checklist. The fidelity checklist included questions which assessed content, coverage, frequency, duration, quality of delivery, and potential moderators. As displayed in (**S1 Appendix**), "content" assessed the key activities that were offered at each school. "Coverage" measured the number of children seen over two grades per school. "Frequency" measured the number of times the hygienist visited the school per year and "duration" the number of hours spent at a school per day by the hygienists. In terms of the moderators measured, quality of delivery referred to the manner in which the program was carried out in terms of preparedness, interaction with teams and communication of activities. Facilitation strategies assessed factors that existed to optimise fidelity such as availability of chair-side assistance, school support and functioning equipment. Participant's responsiveness assessed how the oral hygienists, teachers and parents engaged with the program. Our study also included age and gender as potential moderators as they represent some of the key determinants that influence levels of dental decay among children [23], see **Fig 1**.

Each of the fidelity elements had a different number of items as in (**S1 Appendix**). The scores of content had 10 items and moderator items had 3 items. Coverage, frequency and duration were coded as low "0" or high "1", the benchmarks being 50% for coverage, twice a year for frequency and four hours for duration. This allowed for the coding of the elements as item scores. All the above components were then added to determine an adherence score out of 13. The adherence score of each school program had to be scored out of 13 and a percentage of the score was determined in order to express the level of fidelity, see **Table 1**. The collection of fidelity data was undertaken by the first author and the tools were pre-tested on a school oral health program not involved in the study in order to ensure repeatability [24].

## Ethics

Written informed consent was obtained from the study participants after explaining the purposes of the study. Ethical approval for this study was obtained from the Human Research

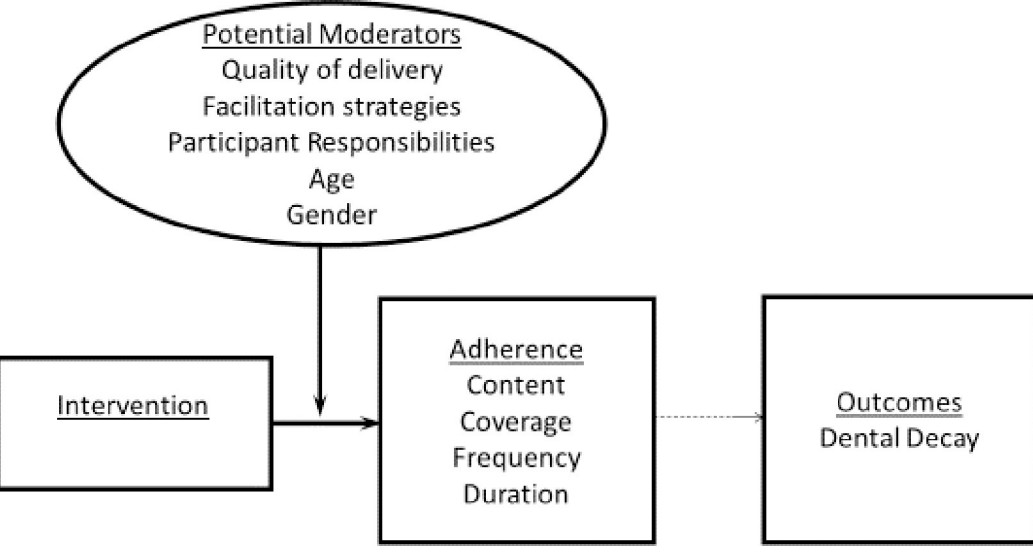

**Fig 1. Modified framework adapted from Carrol et al, 2007.**

**Table 1. Description of services, levels of fidelity and dental scores at school level.**

| Schools | Number of learners seen/total of grades 1 & 7 per school | Proportion of learners covered for 2017 | Content scores/10 & proportions | Fidelity scores/13 & proportions | Decay score (SD) |
|---|---|---|---|---|---|
| 1 | 155/322 | 48% | 2 20% | 3 23% | 0.49 (1.10) |
| 2 | 112/150 | 75% | 3 30% | 6 46% | 0.19 (0.46) |
| 3 | 229/136 | 168% | 4 40% | 8 61% | 0.29 (0.68) |
| 4 | 140/112 | 125% | 3 30% | 6 46% | 0.26 (0.71) |
| 5 | 205/113 | 181% | 4 40% | 7 53% | 0.38 (1.76) |
| 6 | 268/386 | 69% | 4 40% | 5 38% | 1.11 (1.51) |
| 7 | 91/200 | 45% | 1 10% | 2 15% | 0.56 (1.17) |
| 8 | 165/368 | 45% | 1 10% | 2 15% | 1.09 (1.44) |
| 9 | 724/380 | 190% | 3 30% | 7 53% | 0.27 (0.75) |
| 10 | 412/282 | 146% | 1 30% | 6 46% | 0.38 (1.16) |
| Totals & average scores | 2501/2449 | 109.1% | Mean 2.8 (1.0) 28.0% | Mean 5.2 (2.0) 39.6% | Mean 0.54 (1.16) |

Ethics Committee of the University of the Witwatersrand (M170115), and permission to access the schools was granted by the district's Department of Education.

## Data management and analysis

The data from the oral health assessments consisted of continuous variables and that from the fidelity checklist included scores and binary variables. The levels of fidelity at the schools were derived from the adherence scores and were assessed using descriptive statistics, namely proportions. The decay level was assessed using mean scores and standard deviations.

Multilevel Poisson's regression analysis was then utilised to assess the predictive elements of fidelity on the risks of decay. Variables from the model were then fitted on a predictive margins plot in order to get an illustration of how fidelity related to decay. For the regression analysis, fidelity scores were categorised as low when they were less than 50% and high when over 50% [16]. A simple regression analysis was also undertaken to determine the minimum amount of fidelity required for zero levels of dental caries.

For assessing the moderating factors, interaction effects had to be added to the multilevel Poisson regression model. This was in order to assess which of the moderating variables interacted with fidelity in influencing the dental decay outcome. Variables that had statistically significant interactions with fidelity were identified as moderators of fidelity. Quality of delivery and participant responsiveness were omitted from the model due to elements being collinear with fidelity. All the analysis was conducted on STATA version 16 at a 95% confidence interval.

## Results

There were 736 pupils who participated in an oral health exam with ages ranging between 6 and 16 years old. Across the 10 schools involved in the study, only 28% of the program content was covered, however the average percentage of pupils exposed to the program in 2017 was high (109%). The elements of content that were adhered to mostly included brushing and fissure sealant activities (**S1 Appendix**). Adherence was poor on community and resource assessments, training, advocating for dietary practices and collective implementation (**S1 Appendix**). The hygienist spent an average of 3.5 hours at a school per morning over 4 weeks; 50% of the hygienist revisited the school twice a year and the other 50% revisited once a year. As seen in **Table 1,** the lowest level of fidelity was 15%, the highest 61% and the average level

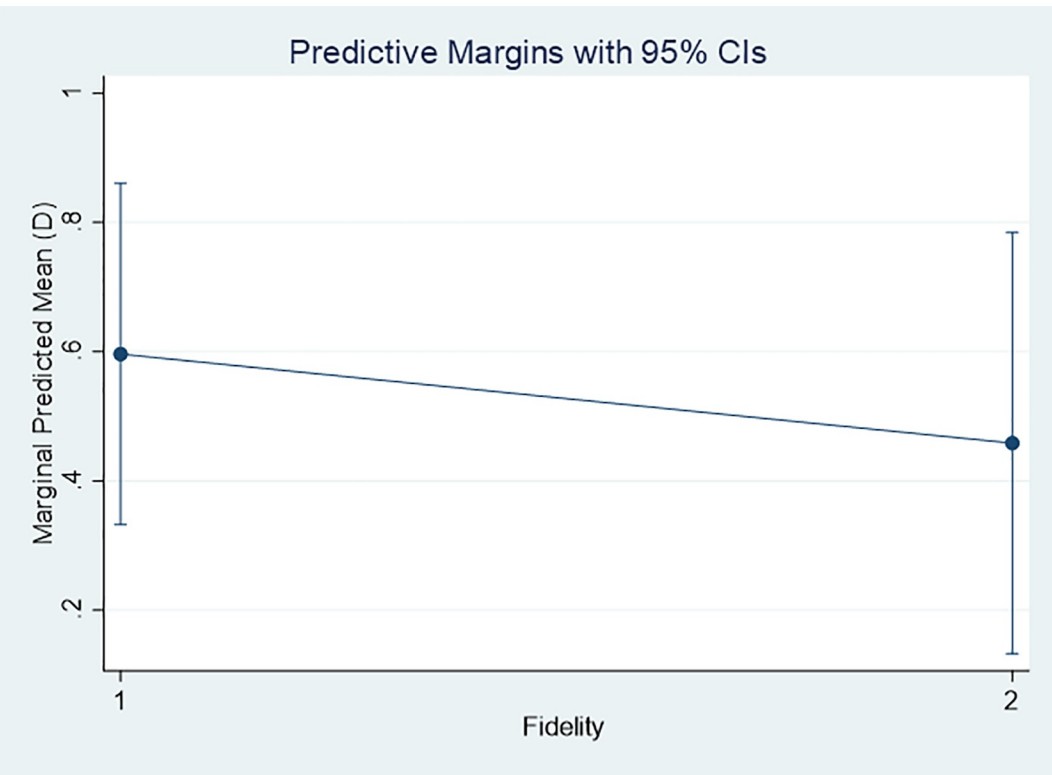

**Fig 2. Margin plot demonstrating relationship between fidelity and mean decay scores.**

of fidelity achieved across the schools was 39.6%. The mean decay (D) score of all the pupils was (0.54; SD: 1.16). School number 2 had the lowest level of decay (0.19; SD: 0.46) while School number 6 had the highest level of decay (1.11; SD: 1.51).

Upon fitting the variables of fidelity and the decay scores on a margins plot, the figure derived displayed how increasing levels of fidelity resulted in a decrease in the mean dental decay across the 10 schools, see **Fig 2.** In addition the simple regression analysis indicated that a minimum fidelity level of 79.8% was required for an ideal zero dental caries score.

**Table 2** demonstrates how the adherence elements of fidelity influence the decay experience after adjusting for age and gender. The Incidence Rate Ratio (IRR) was computed to measure the relative measure of risk of the outcome. The results therefore highlighted that the relative risk of duration, coverage, content and age were found to significantly predict decay

**Table 2. Relationship of fidelity elements with decay experience.**

| D | IRR | Std. Err | p | CI |
|---|---|---|---|---|
| Duration | 0.49 | 0.15 | 0.02 ** | 0.26–0.90 |
| Coverage | 0.54 | 0.12 | 0.008** | 0.34–0.85 |
| Frequency | 0.66 | 0.21 | 0.21 | 0.34–1.27 |
| Content | 1.36 | 0.20 | 0.03** | 1.01–1.81 |
| Age | 2.14 | 0.25 | 0.00** | 1.70–2.69 |
| Gender | 1.05 | 0.10 | 0.60 | 0.86–1.28 |

**p$\leq$ 0.05.

**Table 3. Moderating effects of fidelity on dental decay.**

| D | IRR | Std. Err | p | CI |
|---|---|---|---|---|
| Services offered*Fidelity | 3.45 | 2.84 | 0.13 | 0.68–17.32 |
| Quality of delivery*Fidelity | omitted | | | |
| Facilitation strategy*Fidelity | | | | |
| *Chairside Assistance* | 0.08 | 0.06 | 0.001** | 0.02–0.36 |
| Participant responsiveness*Fidelity | omitted | | | |
| Duration*Fidelity | | | | |
| *4hours* | 0.17 | 0.13 | 0.02** | 0.03–0.80 |
| Age*Fidelity | | | | |
| *8–16* | 0.52 | 0.12 | 0.008** | 0.32–0.84 |
| Gender*Fidelity | | | | |
| *Male* | 0.82 | 0.18 | 0.39 | 0.53–1.27 |

**p≤ 0.05.

levels. Duration (IRR: 0.49;p = 0.02) and coverage (IRR:0.54;p = 008) reduced the risk of decay, while content (IRR:1.36; p = 0.03) and age (IRR:2.14; p = 0.00) increased the risk of decay respectively.

Variables that were found to have moderating effects on fidelity included facilitation strategy, duration and age. When facilitation strategy interacted with fidelity, particularly the presence of a chairside assistant; the relative risk of caries is reduced by 92%, (IRR: 0.08; p = 0.001). Furthermore when duration of the activities was a minimum of 4 hours and learners with ages between 8–16 interacted with fidelity, they reduced the relative risks of decay by 83%, (IRR:0.17;p = 0.02) and 48%, (IRR:0.52;p = 0.008), respectively. **See Table 3.**

## Discussion

The oral hygienists covered about a quarter (28%) of the expected program content, however the overall coverage of the program was high (109%). The average level of fidelity obtained was 40% and was shown to be inversely correlated with the proportion of children with decay, as decrease in levels of decay was associated with increase in the level of fidelity. Fidelity elements that were found to be direct predictors of decay were program duration, its coverage, content and age of the children. The moderating factors of fidelity which indirectly influenced the levels of decay included facilitation strategy, duration and age.

The 40% level of fidelity in this study was similar to those reported elsewhere [15, 24]. Our results also demonstrated that to attain the ideal of zero levels of decay, an 80% level of fidelity would be required. Furthermore, our observed low fidelity of 40% imply that much more effort needs to be made to improve the quality of implementation in addressing the high burden of untreated decay (89.6%) among affected primary school children [25].

Elements that were associated with reduced decay included duration and coverage. The average duration spent at a school was 3.5 hours per session and half of the schools received one follow-up visit a year, yet their coverage in terms of number of children seen, was high. This meant that one contact time was spent per child in a year, and more children beyond the targeted grades of 1 and 7 were reached. It can therefore be assumed that the many children covered were attended to quickly as the content achieved was low (28%) in the single contact per year. A study in Kuwait demonstrated how the synergy between duration and coverage effectively reduced the need for restorative work in school services, as the oral health

practitioners involved spent a lot more time covering large numbers of children (80%) and offering preventative activities without compromising on quality [26].

One of the predictors of increasing levels of decay was age (IRR:2.24). The effect of age was expected, particularly among children between the ages of 8–16. As children get older and transition to the adolescent stages, their oral health knowledge, attitude and practices tends to be low [13]. In addition they often opt for sugary snacks and sugar sweetened beverages which contribute to dental caries [27]. Our results further demonstrated that as age interacted with fidelity, it significantly reduced the relative risk by 48%, from (2.14) to (0.52). This finding also implies that fidelity can moderate the effect of age on dental caries by reducing the risk of decay [17].

Content was also shown to be associated with an increase in levels of decay (IRR:1.36). We also noted the worst performing content elements to include items such as, community resource assessments, advocating for dietary health practices and multi-sectoral collaboration (see **S1 Appendix**). These have been cited as some of the key elements of success in oral health programs [8, 12]. In our study, high content was associated with increased levels of decay possibly because of the limited time given to the elements of program content. Therefore, a high content within the limited time was likely to affect the quality of implementation and clinical outcomes [28].

The elements that were found to moderate and indirectly influence the levels of decay included the facilitation strategy, duration and age. These elements interacted with fidelity to buffer the risk of dental decay. Among all the facilitation strategies assessed, the availability of a dental assistant superseded the other factors and was predicted to reduce decay by 92%. This may be because dental procedures such as fissure sealants require extra precautionary measures in creating a moisture free environment for effective longevity of the restorations, hence the need for chair side assistance [29]. Thought should be given to offering alternative similar effective preventative procedures that do not require a dental assistance, such as fluoride varnishing or rinsing activities [30].

When duration interacted with fidelity it influenced the outcomes of dental caries by further reducing the relative risk by 83%. This therefore emphasised the need for the oral hygienist to spend more hours at the school not only for carrying out preventative techniques, but also for other activities such as thorough planning, training of educators and garnering support from key stakeholders such as parents and community health workers [3]. Community health workers are lay health workers in South Africa who are deployed to improve community access to primary health care services by engaging closely with communities, linking communities to healthcare and providing health education and promotional activities [31]. These community health workers have been effectively used to support HIV programmes and management of childhood illness in South Africa [32]. However their potential support in oral health has not been explored.

## Conclusions and recommendations

The extent of implementation fidelity was 40% and adherence elements which were shown to reduce the risk of decay included duration and coverage. Those that increased the risk of decay were content and age. Potential moderators included facilitation strategies, duration and age; these were predicted to reduce the risk of decay.

The implications of this study are that, in as much as the hygienists covered a high number of children for the purposes of meeting their annual performance targets, they compromised on the program content and levels of fidelity [33]. This negatively impacts on the effectiveness of school programs in controlling the decay burden [6]. Increasing the duration contact time

at the schools also emerged as a key for reducing decay; therefore more time needs to be invested in the carrying out the program at the school sites. Additional support for the oral hygienists at school sites is therefore required for them to effectively carry out their work. Thus the potential for training community health workers in oral health education should be explored as they can assist teachers in providing oral health education and facilitating group tooth brushing activities. This would allow the oral hygienists to spend longer focused time on fissure sealants, curative care, and program reinforcement. The increased duration on the program will improve fidelity and hence quality [34].

Process mapping tools and checklists of the program content needs to be introduced to the oral hygienists in order that they have a clear understanding of the key components of the program and be aware of the necessary activities they need to attend to for achieving their intended outcomes [35]. This would assist the hygienist in monitoring themselves and additionally provide guidance towards improving the delivery of the program content. Improving adherence to the key program components would subsequently improve fidelity and the outcomes of decay [19].

While our study was not designed as a controlled experiment and could not validate the efficacy of the program, we were able to understand the extent of implementation of the program elements. In addition the predictive regression analysis was used in order to predict effects of the direct and indirect associations between fidelity and the levels of decay among the children. This information will be useful for future program planning. The elements of fidelity identified to influence the risks of decay reemphasize the need to invest more time and human capacity into the program.

## Supporting information

**S1 Appendix. Presentation of fidelity elements and how they were measured.**
(DOCX)

**S1 Dataset.**
(XLS)

## Acknowledgments

I acknowledge all the managers and oral hygienists in Gauteng who participated in the study, thank you.

## Author Contributions

**Conceptualization:** Mpho Molete, Aimee Stewart, Jude Igumbor.

**Formal analysis:** Mpho Molete, Jude Igumbor.

**Funding acquisition:** Mpho Molete.

**Investigation:** Mpho Molete.

**Methodology:** Jude Igumbor.

**Project administration:** Mpho Molete.

**Supervision:** Aimee Stewart, Jude Igumbor.

**Validation:** Jude Igumbor.

**Visualization:** Aimee Stewart.

**Writing – original draft:** Mpho Molete.

**Writing – review & editing:** Mpho Molete, Aimee Stewart, Jude Igumbor.

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
