## [Decision Letter · Decision Letter 0]

30 Sep 2020

PONE-D-20-08525

Implementation Fidelity of School Oral Health Programs at a District in South Africa

PLOS ONE

Dear Dr. Molete,

Thank you for submitting your manuscript to PLOS ONE. After careful consideration, we feel that it has merit but does not fully meet PLOS ONE’s publication criteria as it currently stands. Therefore, we invite you to submit a revised version of the manuscript that addresses the points raised during the review process.

We look forward to receiving your revised manuscript.

Kind regards,

Frédéric Denis, Ph.D.

Academic Editor

PLOS ONE

Journal Requirements:

2. Please amend your list of authors on the manuscript to ensure that each author is linked to an affiliation. Authors’ affiliations should reflect the institution where the work was done (if authors moved subsequently, you can also list the new affiliation stating “current affiliation:….” as necessary).

Reviewers' comments:

Reviewer's Responses to Questions

**Comments to the Author**

1. Is the manuscript technically sound, and do the data support the conclusions?

Reviewer #1: Yes

2. Has the statistical analysis been performed appropriately and rigorously? 

Reviewer #1: Yes

3. Have the authors made all data underlying the findings in their manuscript fully available?

Reviewer #1: No

4. Is the manuscript presented in an intelligible fashion and written in standard English?

Reviewer #1: Yes

5. Review Comments to the Author

Reviewer #1: The article is well written.

#1 - The study measures fidelity of implementation. However, it is never made clear exactly what is being implemented. What are the components of the school dental health program? This information could be provided in the Background section, or in the Methods section, or perhaps in a table.

1. Who are the dental hygienists? What is their educational background, and what training do they receive to implement the school dental health program?

2. How many schools do they visit in a day or in a week? How long are they in each school?

3. How many students do they need to see during a visit to a school?

4. What are all the activities that are to carry out during a visit to a school? Do their activities include promotion / education about dental hygiene? Do they talk to groups or classes of children about dental hygiene?

5. What is the role, if any, of school staff such as teachers and principals in implementation of the school dental health program?

#2 - What Department or Ministry in the Government is responsible for the program? Do they carry out any routine monitoring of implementation of the program?

#3 - The conclusions / recommendations could be strengthened. How can implementation of the program be improved?

6. PLOS authors have the option to publish the peer review history of their article (what does this mean?). If published, this will include your full peer review and any attached files.

Reviewer #1: **Yes: **Peter Winch

---

## [Author Response · Author response to Decision Letter 0]

23 Oct 2020

Dear Reviewers, thank you for the useful feedback. I have responded to all the comments as carefully as I could. See the response reviewer letter for more information.

---

## [Editor Report · Decision Letter 1]

26 Oct 2020

Implementation Fidelity of School Oral Health Programs at a District in South Africa

PONE-D-20-08525R1

Dear Dr. Molete,

We’re pleased to inform you that your manuscript has been judged scientifically suitable for publication and will be formally accepted for publication once it meets all outstanding technical requirements.

Kind regards,

Frédéric Denis, Ph.D.

Academic Editor

PLOS ONE
---

## [Editor Report · Acceptance letter]

3 Nov 2020

PONE-D-20-08525R1 

Implementation fidelity of School Oral Health Programs at a District in South Africa. 

Dear Dr. Molete:

I'm pleased to inform you that your manuscript has been deemed suitable for publication in PLOS ONE. Congratulations! Your manuscript is now with our production department. 

Kind regards, 

on behalf of

Dr. Frédéric Denis 

Academic Editor

PLOS ONE